# Calibration of a Catadioptric System and 3D Reconstruction Based on Surface Structured Light

**DOI:** 10.3390/s22197385

**Published:** 2022-09-28

**Authors:** Zhenghai Lu, Yaowen Lv, Zhiqing Ai, Ke Suo, Xuanrui Gong, Yuxuan Wang

**Affiliations:** School of Opto-Electronic Engineering, Changchun University of Science and Technology, Changchun 130022, China

**Keywords:** machine vision, catadioptric camera, projector, system calibration, 3D reconstruction

## Abstract

In response to the problem of the small field of vision in 3D reconstruction, a 3D reconstruction system based on a catadioptric camera and projector was built by introducing a traditional camera to calibrate the catadioptric camera and projector system. Firstly, the intrinsic parameters of the camera and the traditional camera are calibrated separately. Then, the calibration of the projection system is accomplished by the traditional camera. Secondly, the coordinate system is introduced to calculate, respectively, the position of the catadioptric camera and projector in the coordinate system, and the position relationship between the coordinate systems of the catadioptric camera and the projector is obtained. Finally, the projector is used to project the structured light fringe to realize the reconstruction using a catadioptric camera. The experimental results show that the reconstruction error is 0.75 mm and the relative error is 0.0068 for a target of about 1 m. The calibration method and reconstruction method proposed in this paper can guarantee the ideal geometric reconstruction accuracy.

## 1. Introduction

In recent years, catadioptric cameras have been widely used in medical imaging [1,2,3], augmented reality [4,5,6], virtual reality [7,8,9], and three-dimensional (3D) measurement [10,11] or reconstruction [12,13] because of their large field of view, simple structure, rotational symmetry, and high performance-to-price ratio. In the aspect of 3D reconstruction or measurement, based on structured light, 3D reconstruction is an active triangulation technology, which obtains the corresponding relationship between camera image points and projection pattern points through the coding and decoding of structured light. Then, the three-dimensional coordinates of the surface points of the object are obtained by the triangulation principle. The system consists mainly of a camera, a projector, and a computer. Three-dimensional reconstruction based on structured light using a catadioptric camera consists of structural design, system calibration, structured light coding and decoding, and a reconstruction algorithm. After the structure of the system is determined, it is necessary to calibrate the system parameters to obtain the intrinsic and extrinsic parameters of the system. The camera obtains the projection information of the coded structured light combined with the system parameters, and the reconstruction algorithm is used to complete the 3D reconstruction. The calibration of the intrinsic and extrinsic parameters of the system is an important step in 3D reconstruction. There is much research on the individual calibration of catadioptric cameras, such as Lin H et al. [14], who used the unified sphere model to calibrate a single-viewpoint catadioptric camera, and Guo X et al. [15], who used the Taylor polynomial model to calibrate the catadioptric camera. The active Vision Research Group of Oxford University released the widely used calibration toolbox [16] for the calibration of catadioptric cameras. For the problem of projector calibration, it is generally combined with ordinary cameras into a 3D reconstruction system for calibration. For example, Gao Y et al. [17] used the calibration method based on Gray-code and used the camera to calibrate the projector. Jin Y et al. [18] fixed the camera to calibrate the hollow ring, changed the position of the projector, projected the horizontal and vertical stripes, and calibrated the projector with phase equivalence.

For projector and catadioptric camera calibration problems in the system, Jia T et al. [19] used two guide rails to realize the movement of four projectors, which translated into a projector calibration method with multiple reference surfaces, where the projectors projected coded structured light and thus calculated the extrinsic parameters of the catadioptric camera and projector. This system required precision guideways, high computational effort, and difficult data processing. Cordova-Esparza et al. [20] calibrated the extrinsic parameters of a catadioptric camera and a catadioptric projector with the three-times calibration method. Firstly, the two mirrors were removed, the semi-transparent calibration plate between the camera and the projector was pre-calibrated, and then the mirrors were installed and the extrinsic parameters of the catadioptric camera and the catadioptric projector were calibrated using the checkerboard projection of the projector. This method had the disadvantages of intricate debugging, installation and disassembly operations, large accumulated errors, and large data processing.

Coding patterns based on triangulation include the De Bruijn sequence and improved De Bruijn sequence [21]. There are also the n-step phase shift method [22] and the multi-frequency heterodyne method based on the phase height method [23]. Nevertheless, there are a few types of structured light used for 3D reconstruction or measurement of catadioptric systems, such as hourglass structured light [19] and stereo point spatial coding [20].

Therefore, in this paper, a calibration method for a catadioptric system based on structured light is proposed with the help of traditional cameras. First, the projector is calibrated, then the catadioptric camera is calibrated. The extrinsic parameters of the catadioptric camera are calibrated by shooting the same calibration plate with the auxiliary camera and the catadioptric camera. This method makes full use of the existing calibration toolbox and can quickly and easily calibrate the whole system. Because of the uniform sphere model used by the catadioptric camera, a 3D space-point coordinate formula based on the uniform sphere model is derived in this study. The reconstruction of the white reflective geometry also shows the measures taken to decode the effect of structured light before and after the effect. Finally, binary time-coded structured light is used to reconstruct the object with an error of 0.75 mm and a relative error of 0.068.

This paper is arranged as follows: in the second part, the basic model of the whole system is provided, including the catadioptric camera model, pinhole imaging model, and projector model; in the third part, how to use the traditional auxiliary camera to calibrate the whole system is depicted in detail, and the 3D reconstruction equation of surface structured light is derived from the spherical unified model and the projector model. Subsequently, in the fourth part, the whole experimental system is constructed, the whole system is calibrated by this method, the 3D reconstruction of the object is completed, and the experimental results are analyzed and discussed. Finally, a summary and outlook of the whole article are given.

## 2. Materials and Methods

### 2.1. Related Basic Knowledge

#### 2.1.1. Catadioptric Camera Model

The widely used unified spherical imaging model [24] was selected to describe the imaging process of the catadioptric camera. The schematic diagram of the unified sphere model is shown in Figure 1. In Figure 1, we can clearly see the process of projecting a point $P_{C}(X_{C},Y_{C},Z_{C})$ in the camera coordinate system to the image pixel p(u,v). The mapping relationship between the 3D spatial point $P_{C}$ in the camera coordinate system and the image pixel p is as follows:(1)$$p=\left[\begin{array}{c} u\\ v\\ 1 \end{array}\right]=M_{O}h(P_{C})=M_{O}p_{U}=\left[\begin{array}{ccc} \gamma_{1} & \gamma_{1}\alpha & u_{0}\\ 0 & \gamma_{2} & v_{0}\\ 0 & 0 & 1 \end{array}\right]\left[\begin{array}{c} \frac{X_{S}}{Z_{S}+\xi }\\ \frac{Y_{S}}{Z_{S}+\xi }\\ 1 \end{array}\right]$$
where $M_{O}$ is the intrinsic parameter matrix of the catadioptric camera, including focal length $\left(\gamma_{1},\gamma_{2}\right)$, deflection angle α (usually 0), and principal point ($u_{0},v_{0}$). The point $p_{U}(u_{S},v_{S})$ is on the normalized plane. The unit sphere is translated ξ unit length and normalized to obtain the normalized plane. $P_{S}(X_{S},Y_{S},Z_{S})$ is the projection coordinate of point $P_{C}$ on the unit sphere, and the relation is:(2)$$\left[\begin{array}{c} X_{S}\\ Y_{S}\\ Z_{S} \end{array}\right]=\left[\begin{array}{c} X_{C}\\ Y_{C}\\ Z_{C} \end{array}\right]/\sqrt{X_{C}{}^{2}+Y_{C}{}^{2}+Z_{C}{}^{2}}$$

In the unified spherical model and the pinhole camera model [25], the relationship between the world coordinate system and the camera coordinate system is consistent. In addition, the point $P_{C}$ in the camera coordinate system is obtained through the spatial transformation of point $P_{W}$ in the world coordinate system, and its corresponding relation is as follows:(3)$$\left[\begin{array}{c} X_{C}\\ Y_{C}\\ Z_{C}\\ 1 \end{array}\right]=\left[\begin{array}{cc} R & t\\ \vec{0} & 1 \end{array}\right]\left[\begin{array}{c} X_{W}\\ Y_{W}\\ Z_{W}\\ 1 \end{array}\right]=M_{O}P_{W}$$
where $M_{O}$ is the extrinsic parameter matrix of the catadioptric camera, including the rotation matrix R and the translation matrix t.

By calibrating the camera, the intrinsic and extrinsic parameters of the camera are obtained, and then the projection points $P_{S}$ on the unit sphere corresponding to the image points p are calculated by using these parameters. The equation between the two is as follows:(4)$$\left[\begin{array}{c} X_{S}\\ Y_{S}\\ Z_{S} \end{array}\right]=\left[\begin{array}{c} \frac{\xi +\sqrt{1+\left(1-\xi^{2}\right)\left(u_{S}{}^{2}+v_{S}{}^{2}\right)}}{u_{S}{}^{2}+v_{S}{}^{2}+1}u_{S}\\ \frac{\xi +\sqrt{1+\left(1-\xi^{2}\right)\left(u_{S}{}^{2}+v_{S}{}^{2}\right)}}{u_{S}{}^{2}+v_{S}{}^{2}+1}v_{S}\\ \frac{\xi +\sqrt{1+\left(1-\xi^{2}\right)\left(u_{S}{}^{2}+y^{2}\right)}}{u_{S}{}^{2}+v_{S}{}^{2}+1}-\xi \end{array}\right]$$
where $\left(u_{S},v_{S}\right)$ are the coordinates of points on the normalized plane, calculated as:(5)$$\left[\begin{array}{c} u_{S}\\ v_{S}\\ 1 \end{array}\right]=K^{-1}p$$

In order to describe the nonlinear relationship caused by lens distortion more accurately, the parameters of radial distortion $k_{1}$, $k_{2}$, $k_{3}$ and tangential distortion $p_{1}$, $p_{2}$ are introduced. The equation of points before and after the correction of distortion is:(6)$$\{\begin{array}{c} x_{\mathrm{distorted}}=x\left(1+k_{1}r^{2}+k_{2}r^{4}+k_{3}r^{6}\right)+2p_{1}xy+p_{2}\left(r^{2}+2x^{2}\right)\\ y_{\mathrm{distorted}}=y\left(1+k_{1}r^{2}+k_{2}r^{4}+k_{3}r^{6}\right)+p_{1}\left(r^{2}+2y^{2}\right)+2p_{2}xy \end{array}$$
where (x,y) is the distortion of coordinates in the image coordinate system and r is the distance from (x,y) to the origin of the image coordinate system.

#### 2.1.2. Pinhole Imaging Model of the Traditional Camera

There are many models to describe traditional cameras, and the model adopted by many people is the pinhole imaging model [25]. Figure 2 displays the schematic diagram of the pinhole imaging model. It furthermore explains how the point $P_{C}$ in the camera coordinate system is projected to the physical image plane through the camera aperture, and finally, the corresponding pixel coordinate p is obtained. Under the pinhole imaging model, the relationship between a point $P_{W}$ in space and its corresponding image point p is:(7)$$Z_{C}\left[\begin{array}{c} u\\ v\\ 1 \end{array}\right]=\left[\begin{array}{cc} \begin{array}{ccc} f_{x} & 0 & u_{0}\\ 0 & f_{y} & v_{0}\\ 0 & 0 & 1 \end{array} & \begin{array}{c} 0\\ 0\\ 0 \end{array} \end{array}\right]\left[\begin{array}{cc} R & t\\ \vec{0} & 1 \end{array}\right]\left[\begin{array}{c} X_{W}\\ Y_{W}\\ Z_{W}\\ 1 \end{array}\right]=M_{C}A_{C}P_{W}$$
where $M_{C}$ is the intrinsic parameter matrix of the camera, and parameter $f_{x},f_{y},u_{0},v_{0}$ is determined by the manufacturing process of the camera. Matrix $A_{C}$ represents the position relationship between the camera and the world coordinate system, also known as the extrinsic parameter matrix of the camera.

In the pinhole imaging model, there is also nonlinear distortion caused by lens refraction. The distortion correction model is the same as that of the catadioptric camera, which consists of radial distortion and tangential distortion.

#### 2.1.3. Projector Projection Model

The projector is a projection device used to enlarge the display image. Through aperture imaging, the image plane is projected into space. The curtain or wall board is used to accept the image in the space to achieve the image magnification and projection. The imaging model is similar to the pinhole model [26].

Point p(u,v) in the pixel coordinate system of the projector image can be transformed by the matrix to point $P_{W}\left(X_{W},Y_{W},Y_{W}\right)$ in the corresponding world coordinate system. The coordinate transformation equation of this process can be expressed as:(8)$$Z_{P}\left[\begin{array}{c} u_{P}\\ v_{P}\\ 1 \end{array}\right]=\left[\begin{array}{cc} \begin{array}{ccc} f_{x} & 0 & u_{0}\\ 0 & f_{y} & v_{0}\\ 0 & 0 & 0 \end{array} & \begin{array}{c} 0\\ 0\\ 0 \end{array} \end{array}\right]\left[\begin{array}{cc} R_{P} & t_{P}\\ \vec{0} & 1 \end{array}\right]\left[\begin{array}{c} \begin{array}{c} X_{W}\\ Y_{W}\\ Y_{W} \end{array}\\ 1 \end{array}\right]=M_{p}A_{P}P_{W}$$
where $M_{P}$ is the intrinsic parameter matrix of the projector and $A_{P}$ is the intrinsic parameter matrix.

### 2.2. System Calibration and 3D Reconstruction

#### 2.2.1. Calibration of Ordinary Camera and Projector

The calibration of ordinary cameras and projectors is to obtain $M_{C}$, $A_{C}$ in Equation (7) and $M_{P}$, $A_{P}$ in Equation (8), as well as distortion parameters.

The intrinsic and extrinsic parameters of the camera and projector were calculated by Falcao et al. [27]. Foremost, the calibration pattern was pasted on the whiteboard and the checkerboard was projected onto the calibration board. Next, the angle of the calibration plate was altered and the camera was used in turn. Afterwards, the corners in the calibration pattern in the image were detected and the camera was calibrated by Zhang’s method. According to the extrinsic parameters of the camera, when the calibration plate is at any position, the calibration plane equation is calculated. Then. the line between the camera origin and the corner of the projection checkerboard and the calibration plane was established to calculate the 3D coordinates of the feature points, and then the feature points of the image were detected. Ultimately, the corresponding relationship between 2D and 3D projection points was used to calibrate the projector. The calibration flow chart is shown in Figure 3.

#### 2.2.2. Calibration of Catadioptric Camera

Figure 4 shows the calibration process of a catadioptric camera. Due to the large field angle of the catadioptric camera, a large amount of information is contained in a picture, so it is essential to make the side length of the checkerboard larger and the calibration board closer to the catadioptric camera to facilitate the extraction of feature points. With the help of the calibration toolbox [28], the intrinsic parameters of the catadioptric camera are calibrated with a planar chessboard, the calibration plate is folded around the catadioptric camera, and the pictures used for calibration are taken as shown in Figure 4b.

Firstly, initializing the parameters, let $k_{1}\approx k_{2}\approx k_{3}\approx k_{4}\approx k_{5}\approx \alpha \approx 0$ and $\gamma_{1}\approx \gamma_{2}$, as shown in Figure 4b for manual detection (solid red line) and Hough circle detection (dashed blue line). This is used to determine the principal point. Secondly, Figure 4c shows at least 3 points marked horizontally for estimating focal length. In Figure 4d, the feature point regions of all the catadioptric images are pointed out, and the 2D coordinate system and 3D coordinate system of the calibration plate are established. In Figure 4e, all the positions of feature points are detected, and a series of two-dimensional and three-dimensional coordinates of feature points are used to calibrate the catadioptric camera, remove the pictures with large reprojection errors, and finally obtain the internal and external parameters and distortion coefficients of the catadioptric camera.

The reprojection error is often used to describe and calibrate the accuracy of the camera. This method will generate error accumulation and the projector error will be slightly increased. After calibration according to the above steps, Figure 5 is obtained. Figure 5 shows the reprojection error in the X direction and Y direction, which are 0.49 pixels and 0.37 pixels, respectively.

#### 2.2.3. Position and Attitude Calculation of the Projector and Catadioptric Camera

When the extrinsic parameters are calculated by using the Jacobian matrix of the minimized residual function, the rotation matrix and translation vector expressed by quaternion $Q_{w}$ are obtained, and the rotation matrix can be obtained by Equations (9) and (10):(9)$$Q^{\prime }=\frac{Q_{w}}{{\left\|Qw\right\|}_{2}}=\left[\begin{array}{c} q_{0}\\ \begin{array}{c} q_{1}\\ q_{2}\\ q_{3} \end{array} \end{array}\right]\;$$
(10)$$R=\left[\begin{array}{ccc} q_{0}^{2}+q_{1}^{2}+q_{2}^{2}+q_{3}^{2} & 2\left(-q_{0}q_{3}+q_{1}q_{2}\right) & 2\left(q_{0}q_{2}+q_{1}q_{3}\right)\\ 2\left(q_{0}q_{3}+q_{2}q_{1}\right) & q_{0}^{2}-q_{1}^{2}+q_{2}^{2}-q_{3}^{2} & 2\left(-q_{0}q_{1}+q_{2}q_{3}\right)\\ 2\left(-q_{0}q_{2}+q_{3}q_{1}\right) & 2\left(q_{0}q_{1}+q_{3}q_{2}\right) & q_{0}^{2}-q_{1}^{2}-q_{2}^{2}+q_{3}^{2} \end{array}\right]$$

Fixing the plane chessboard, calibrating the projector, and obtaining the extrinsic parameters between the projector and the world coordinate system of the plane chessboard, $R_{p}$ and $t_{p}$, then the relationship between the projector coordinate system and the catadioptric camera coordinate system is:(11)$$R^{\prime }=RR_{P}{}^{-1}$$
(12)$$t^{\prime }=t-RR_{P}{}^{-1}t_{P}$$

### 2.3. Three-Dimensional Reconstruction of Catadioptric System Based on Surface Structured Light

#### 2.3.1. Depth Estimation

The 3D reconstruction of an object based on surface structured light using a catadioptric camera cannot be achieved directly using the triangulation method. It is necessary to project the catadioptric image onto the unit sphere through the normalized plane and then reconstruct it with the projector by triangulation. So, we can obtain:(13)$$\left[\begin{array}{c} {u^{\prime }}_{C}\\ {v^{\prime }}_{C}\\ 1 \end{array}\right]=\left[\begin{array}{ccc} \gamma_{1} & \gamma_{1}\alpha & u_{0}\\ 0 & \gamma_{2} & v_{0}\\ 0 & 0 & 0 \end{array}\right]\left[\begin{array}{c} X_{S}\\ Y_{S}\\ Z_{S} \end{array}\right]$$

And then we can obtain:(14)$$S_{C}\left[\begin{array}{c} {u^{\prime }}_{C}\\ {v^{\prime }}_{C}\\ 1 \end{array}\right]=\left[\begin{array}{cc} \begin{array}{ccc} a_{11} & a_{12} & a_{13}\\ a_{21} & a_{22} & a_{23}\\ a_{31} & a_{32} & a_{33} \end{array} & \begin{array}{c} a_{14}\\ a_{33}\\ a_{34} \end{array} \end{array}\right]\left[\begin{array}{c} \begin{array}{c} X_{W}\\ Y_{W}\\ Z_{W} \end{array}\\ 1 \end{array}\right]$$
where $S_{C}$ is a scaling factor. According to Section 2.1.3, we can also obtain the coordinate transformation of the projector, which is:(15)$$Z_{P}\left[\begin{array}{c} u_{P}\\ v_{P}\\ 1 \end{array}\right]=\left[\begin{array}{cc} \begin{array}{ccc} b_{11} & b_{12} & b_{13}\\ b_{21} & b_{22} & b_{23}\\ b_{31} & b_{32} & b_{33} \end{array} & \begin{array}{c} b_{14}\\ b_{33}\\ b_{34} \end{array} \end{array}\right]\left[\begin{array}{c} \begin{array}{c} X_{W}\\ Y_{W}\\ Z_{W} \end{array}\\ 1 \end{array}\right]$$

Finally, it is obtained that the three-dimensional coordinate of the object in the world coordinate system is:(16)$$\left[\begin{array}{c} \begin{array}{c} X_{W}\\ Y_{W}\\ Z_{W} \end{array}\\ 1 \end{array}\right]={\left(A^{T}A\right)}^{-1}A^{T}b$$
where:(17)$$A=\left[\begin{array}{ccc} {u^{\prime }}_{C}a_{31}-a_{11} & {u^{\prime }}_{C}a_{32}-a_{11} & {u^{\prime }}_{C}a_{33}-a_{13}\\ {v^{\prime }}_{C}a_{31}-a_{21} & {v^{\prime }}_{C}a_{32}-a_{22} & {v^{\prime }}_{C}a_{33}-a_{23}\\ u_{P}b_{31}-b_{11} & u_{C}b_{32}-b_{12} & u_{C}b_{33}-b_{13}\\ u_{P}b_{31}-b_{21} & u_{C}b_{32}-b_{22} & u_{C}b_{33}-b_{23} \end{array}\right]$$
(18)$$b=\left[\begin{array}{c} a_{14}-{u^{\prime }}_{C}a_{34}\\ a_{24}-{v^{\prime }}_{C}a_{34}\\ b_{14}-u_{P}b_{34}\\ b_{24}-u_{P}b_{34} \end{array}\right]$$

#### 2.3.2. Summary of Calibration and Reconstruction Steps

Figure 6 illustrates the whole 3D reconstruction process.

The system calibration and reconstruction algorithms in this paper are summarized as follows:The calibration toolbox [28] was used to calibrate the camera and projector to obtain the intrinsic parameter *M_P_* of the projector and the intrinsic parameter *M_C_* of the camera;The intrinsic parameters K and mirror parameters ξ of the catadioptric camera are obtained by the calibration toolbox [16] to calibrate the catadioptric camera;Taking the plane checkerboard as the reference plane of the world coordinate system, the extrinsic parameters between the camera and the world coordinate system are calculated as R and t;The extrinsic parameters $R_{P}$ and $t_{p}$ between the projector and the world coordinate system are calculated with a fixed plane checkerboard assisted by an ordinary camera;Move the plane checkerboard, repeat steps 3 and 4, and calculate the extrinsic parameters $R^{\prime }$ and $t^{\prime }$ of the catadioptric camera and projector by using Equations (11) and (12);Take $R^{\prime }$ and $t^{\prime }$ as initial values, optimize and minimize the reprojection error, and obtain the final values of $R^{\prime }$ and $t^{\prime }$;Project the coded structured light, use Equation (17) to calculate the 3D-reconstructed point cloud of the object, and use MeshLab to display the point cloud results.

## 3. Results

### 3.1. Design and Calibration of Catadioptric System Based on Structured Light

As shown in Figure 7, the hardware part of the experimental system consists of a catadioptric camera, a traditional camera, a projector, and a computer system. The left and right sides of the screen are respectively placed by the traditional camera and catadioptric camera while the projector is placed against the screen. The projector uses EPSONCB-X06E with a resolution of 1024 × 768. The reflector of the catadioptric camera is made by Bellissimo. Its model is 0–360 Panoramic Optic and the resolution is 4352 × 3264. The ordinary camera is the rear camera of the Mi 10 phone. The steps in Section 3.1 are used to complete the calibration of the projector and the catadioptric camera. The reprojection error of the projector in the X and Y directions is 0.81 and 0.53, respectively. This method will generate error accumulation and the projector error will be slightly increased. The calibration results are shown in Table 1. After calculation by Equations (11) and (12), the optimized result is that
(19)$$R^{\prime }=\left[\begin{array}{ccc} 0.3679 & 0.0530 & -0.9283\\ 0.9281 & -0.0823 & 0.3630\\ -0.0572 & -0.9952 & -0.0795 \end{array}\right],t^{\prime }=\left[\begin{array}{c} 1007.92\\ -559.33\\ 209.53 \end{array}\right].$$

### 3.2. Three-Dimensional Reconstruction of Catadioptric System Based on Surface Structured Light

The projector projects structured light onto objects in turn, and the back of the object can be non-reflective objects such as walls, whiteboards, and light-absorbing fabric. Then, the captured structured light projection picture is decoded.

Due to the particularity of the catadioptric image, in the process of 3D reconstruction, the images are projected onto the unit sphere, and the edge to the center will be compressed into the unfolding process, so we need to pay more attention to whether the decoding is accurate or not. In order to solve the problem of how to make the decoding more accurate, methods based on various structured lights are also different.

There are many ways to optimize structured light decoding, such as stacking patterns to increase the characteristics of patterns [29,30], considering the characteristics of patterns from multiple angles [31,32], and using deep learning methods [33,34]. We take into account the characteristics of the pattern itself and optimize it in combination with image processing. First of all, for the sequence of structured light, an inverse coding sequence is added to help judge whether a point is projected with bright or dark stripes, and the problem of reflecting the light of objects is initially improved. After the $\left[u_{P},v_{P}\right]$ graph matrix is obtained, a median filter is added. Later, in order to display only the decoded values of the projected position, a mask is built to filter the background. At the same time, it further eliminates the influence caused by the reflection of the object.

Figure 8a shows the results of the direct decoding of binary-encoded structured light in this paper, while Figure 8b shows the decoding results after median filtering and a background pickle operation.

The decoded code value is substituted into Equation (16), the three-dimensional point coordinates of the object are calculated, and the depth map is shown in Figure 9a. In this depth map, however, we can see that the depth value in the upper left corner of the curtain is smaller because the catadioptric image is opposite to the left and the right. Using the software MeshLab, the point cloud diagram is shown in Figure 9b.

In Table 2, we can get that the base edge reconstruction error of the cuboid is 0.75 mm, and the relative error is 0.0068. The height reconstruction errors of the cuboid and cone are 3.85 mm and 5.60 mm, respectively. The reconstruction accuracy of cuboid side length is good, but the height error is large. This may be because the aspect ratio of the catadioptric image is 4:3 and the reprojection error in the X-axis direction is bigger than that in the Y-axis direction. 

In order to verify the experimental results, a comparison with the literature—[19,20,35]—was performed, and Table 3 was obtained.

In addition, due to the different sizes of the objects measured, they are transformed into a reconstruction of the length of 10 mm. Table 4 shows the accuracy comparison of the reconstruction of 10 mm by different algorithms.

Jia et al. [19] projected hourglass-like spatial structure light to reconstruct the depth of the plane, measuring the depth every 10 mm with high measurement accuracy. Innovation was made in the aspect of structural light, but the projection of this sort of structural light is sparse, and the object edge cannot be reconstructed, just like the depth measurement of the plane in that paper. Cordova-Esparza et al. [20] and Cesar-Cruz et al. [35] used nearly the same measurement system with a novel device structure and could reconstruct the inner wall of barrels. Cesar-Cruz’s method adopts the phase shift method for a sphere and trolley, so the reconstruction error is too large and there is no depth map or point cloud map. Another method is to reconstruct the intersecting planes by projecting three-dimensional point-like structured light and calculating the angle. In Cordova-Esparza’s method, the point structured light can only be used to measure the depth of the projected points, and the details of the object are not shown, so as to measure the included angle by coplanar points. The reconstruction effect of this paper is to reconstruct the cuboid, cone, and curtain, and the details and relative positions of objects can be measured. The error of measuring 10 mm is 0.07 mm and the relative error is 0.007, which improves the accuracy and measurement range compared to the methods of Cesar-Cruz and Jia. From the experimental results, the calibration of a catadioptric system with time-coded structured light based on this paper seems to be the optimal and better choice for the 3D reconstruction of static objects.

## 4. Discussion

In order to solve the problem of the small field of view of traditional cameras, a 3D reconstruction system was set up in the laboratory based on a catadioptric camera and an ordinary projector. The intrinsic and extrinsic parameter models and 3D reconstruction calculation methods of the system are provided. Aiming at the problem of system calibration, a calibration method using a traditional camera as an auxiliary is proposed. Then, the unified sphere model of the catadioptric camera and the inverse camera model of the projector are used to derive the calculation equation of the three-dimensional space point coordinates of the projected surface structured light of an object, which establishes the theoretical basis for 3D reconstruction or measurement.

There are not many existing research methods for the calibration of catadioptric systems based on surface structured light. Jia T [19] uses the Taylor polynomial calibration toolbox to calibrate the catadioptric camera and uses accurate guides and projected patterns to calibrate the projector. In Cordova-Esparza’s method [20], the Taylor polynomial calibration toolbox is also adopted to calibrate components in steps. The checkerboard calibration repeated three times is required, and each step involves the installation or disassembly of components. In this paper, the calibration method of the catadioptric system is adopted, and the traditional camera is introduced. The parameters of the catadioptric camera and projector are skillfully related. The procedure is simple, and there is no need to disassemble and install devices in the calibration process. The object of reconstruction in this study is white geometry. Not only the surface needs to be reconstructed, but also the edge information needs to be adequately reconstructed. Consequently, the hourglass spatial coding structured light [19] and the 10 × 6 stereo-point spatial coding structured light [20] are not applicable. In addition, the objects they reconstruct or measure are also planes or angles between planes. The sinusoidal fringes projected by Almaraz-Cabral’s method [35] can reconstruct objects such as the car model and ball, but no point cloud image is generated, and the shapes of the measured objects are very different from the original objects. In addition, the surfaces behind the projected pattern will reflect each other. In this paper, the influence of reflected light is removed by adding a median filter and setting up a background film. In addition, all of them used the Taylor polynomial model to calculate 3D reconstruction point coordinates. In this study, the equation derived by the method of triangulation was used to project the catadioptric image onto the unit sphere, which verifies the feasibility of the unified sphere model for 3D reconstruction. At present, in this paper, the reconstruction accuracy of the catadioptric system based on surface structured light is still unable to catch up with the reconstruction accuracy of the traditional camera. One of the main reasons is the calibration error of the system. When calibrating a catadioptric camera, pictures should be added for calibration. The calibration error of the projector includes the calibration error of the traditional camera. We consider increasing the resolution of the projector to make the calibration error of the projector smaller. The second reason is that the distance of the reconstructed object is related. In the process of calibration and reconstruction, the focal length of the catadioptric camera is fixed. In order to shoot near the calibration plate, the focus is near. The further away the object is, the worse the reconstruction accuracy will be. The improvement method is to bring the reconstructed object closer to the catadioptric camera without affecting the projection of the projector and the calibration of the system.

In the future, for the problem that the field of view of the catadioptric camera cannot be fully utilized, we will consider placing the projector on a turntable or changing the structure of the projector to cover the whole field of view of the catadioptric camera with the projected structured light, so as to complete the 3D reconstruction of the whole field of view.

## Figures and Tables

**Figure 1 sensors-22-07385-f001:**
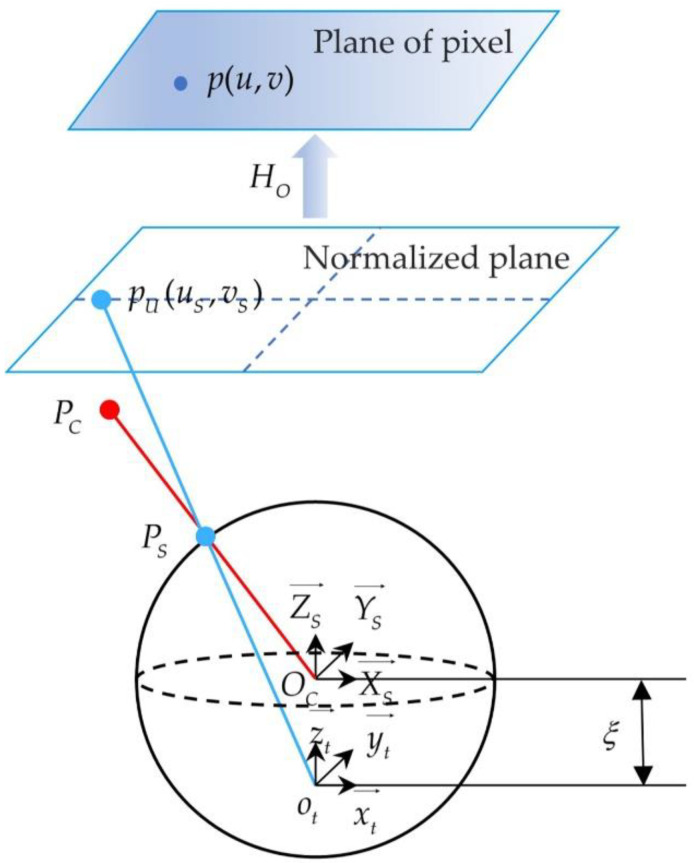
The schematic diagram of the unified sphere model.

**Figure 2 sensors-22-07385-f002:**
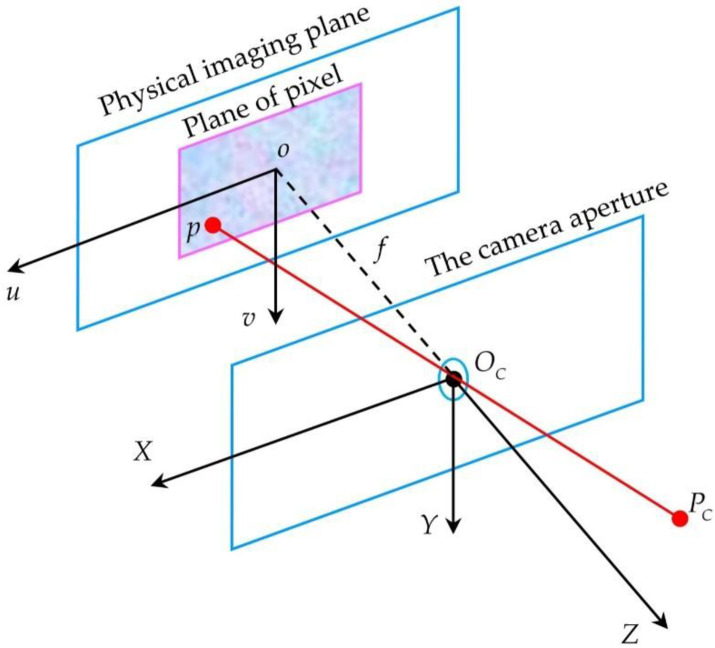
The schematic diagram of the pinhole imaging model.

**Figure 3 sensors-22-07385-f003:**
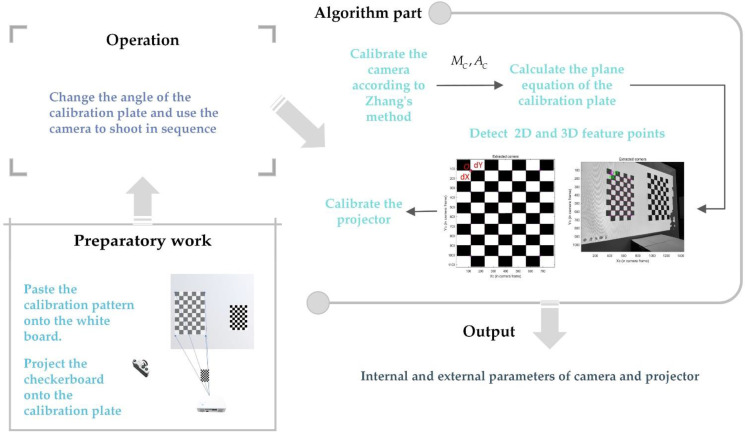
Calibration flow chart of ordinary camera and projector.

**Figure 4 sensors-22-07385-f004:**
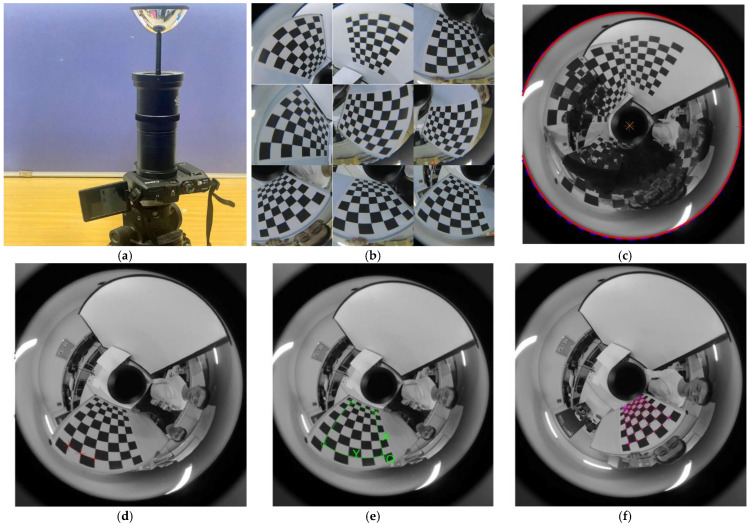
Calibration process of the catadioptric camera. (**a**) The catadioptric camera consists of a reflector and a camera; (**b**) some pictures of the calibration plate were used to calibrate the catadioptric camera; (**c**) principal point estimation; (**d**) the focal length is estimated from three points horizontally; (**e**) framing out the feature point area and establishing the coordinate system; (**f**) detection of all feature points.

**Figure 5 sensors-22-07385-f005:**
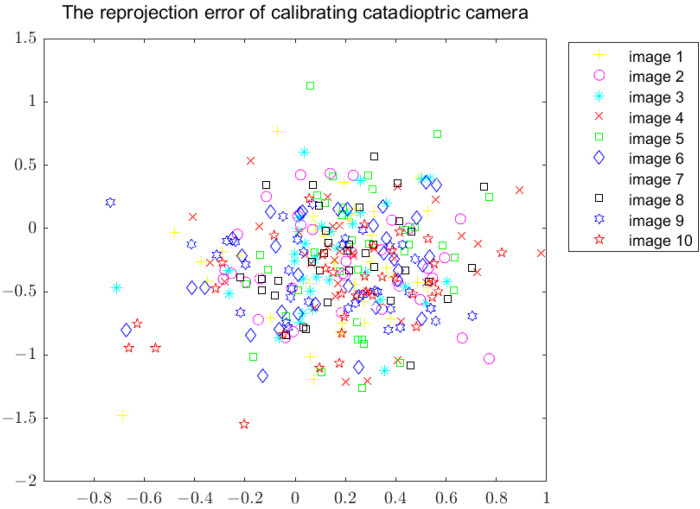
Reprojection error of calibrating the catadioptric camera.

**Figure 6 sensors-22-07385-f006:**
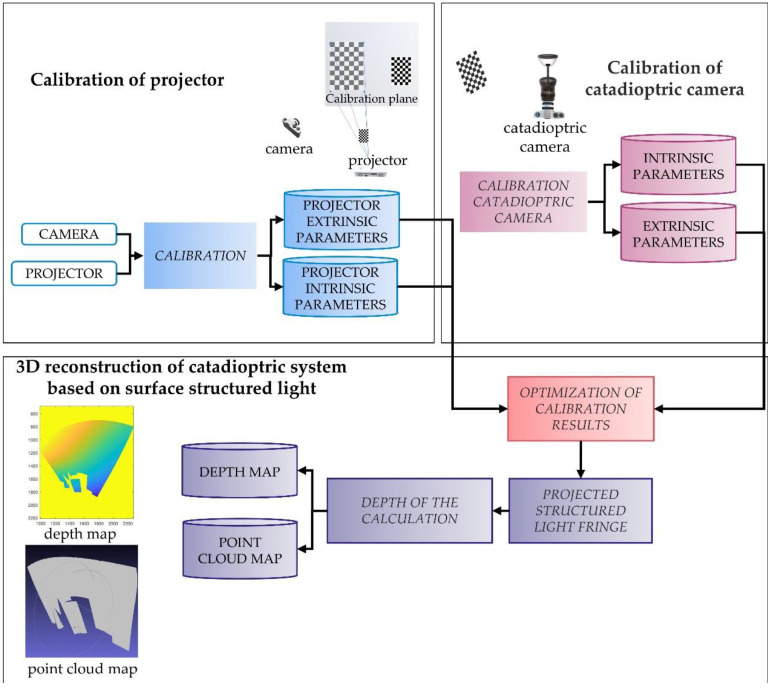
Calibration and 3D reconstruction flow chart of catadioptric system.

**Figure 7 sensors-22-07385-f007:**
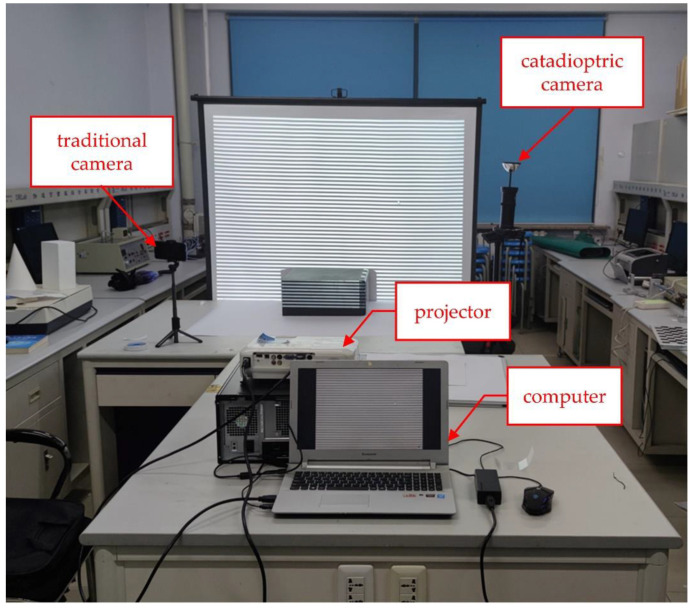
The hardware design of the catadioptric system is based on structured light.

**Figure 8 sensors-22-07385-f008:**
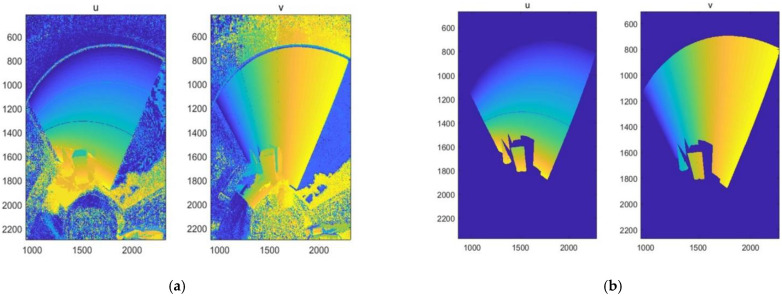
Diagram of structured light decoding. (**a**) The decoding results of the direct decoding of object reflection effects; (**b**) the decoding results of adding median filtering and a background pickle operation to remove the background and reflections.

**Figure 9 sensors-22-07385-f009:**
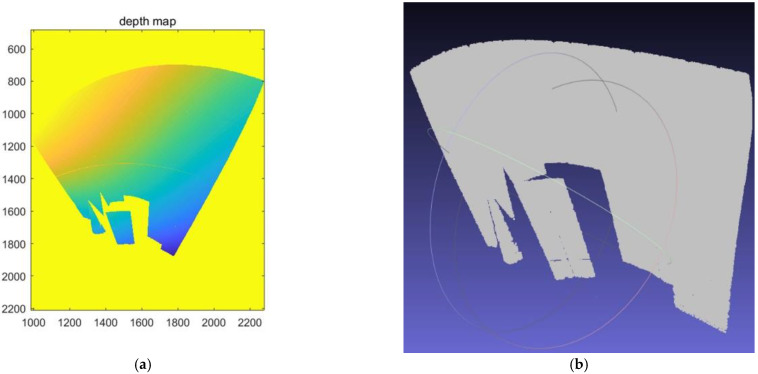
Three-dimensional reconstruction of the geometry. (**a**) Depth map of the projected area; (**b**) using Meshlab software to display the point cloud map of the geometry.

**Table 1 sensors-22-07385-t001:** Intrinsic and extrinsic parameters were obtained by the initial calibration of the catadioptric camera and projector.

Parameters	Catadioptric Camera	Projector
Intrinsic matrix	$\left[\begin{array}{ccc} 932.80 & 0 & 1577.40\\ 0 & 944.38 & 2179.08\\ 0 & 0 & 1 \end{array}\right]$ ξ=0.76934	$\left[\begin{array}{ccc} 4312.98 & 0 & 385.75\\ 0 & 4194.24 & 1252.25\\ 0 & 0 & 1 \end{array}\right]$
Rotation matrix	$\left[\begin{array}{ccc} 0.8182 & -0.0177 & -0.5746\\ 0.5742 & -0.0257 & 0.8183\\ -0.0293 & -0.9995 & -0.0109 \end{array}\right]$	$\left[\begin{array}{ccc} 0.8355 & 0.0267 & 0.5487\\ 0.0253 & 0.9958 & -0.0870\\ -0.5488 & 0.0865 & 0.8314 \end{array}\right]$
Translation matrix	$\left[\begin{array}{c} -36.02\\ -525.47\\ 247.50 \end{array}\right]$	$\left[\begin{array}{c} -354.75\\ -95.91\\ 978.48 \end{array}\right]$

**Table 2 sensors-22-07385-t002:** Comparison of geometry reconstruction values with truth values.

Measurement Object	Parameter	Length (mm)	Height (mm)
Cuboid	Standard parameter	110	220
	Reconstruction parameter	110.75	216.15
	Relative error	0.0068	0.0175
Cone	Standard parameter	140	220
	Reconstruction parameter	140.55	214.40
	Relative error	0.00392	0.0255

**Table 3 sensors-22-07385-t003:** Comparison with the reconstruction methods in the literature: [19,20,35].

Comparative Items	Jia [19]	Cesar-Cruz [20]	Cordova-Esparza [35]	Proposed Method
Camera System	One catadioptric camera and four projectors	One catadioptric camera and a catadioptric projector	One catadioptric camera and a catadioptric projector	One catadioptric camera and a projector
Structured light	Hourglass spatial coding	Temporal phase unwrapping	10 × 6 stereo-point spatial coding	Binary time-coding
Camera projection model	Taylor polynomial model	Taylor polynomial model	Taylor polynomial model	Unified spherical model
The experimental application	Scene depth perception	Three-dimensional reconstruction of the sphere	Measurement of the angle between two planes	Three-dimensional reconstruction of a cuboid

**Table 4 sensors-22-07385-t004:** Measuring 10 mm reconstruction accuracy comparison.

Methods	Measured Value, mm	The Relative Error
Real value	10.00	−
Jia [19]	10.24	0.024
Cesar-Cruz [20]	8.95	0.105
Proposed method	10.07	0.007

## Data Availability

Not applicable.

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
