# Peer review of "Calibration of a Catadioptric System and 3D Reconstruction Based on Surface Structured Light"

_sensors, 2022, doi:10.3390/s22197385_

Round 1
Reviewer 1 Report
In this paper, the authors provide a 3D reconstruction system in the laboratory based on a catadioptric camera and an ordinary projector. The intrinsic and extrinsic parameter models and 3D reconstruction calculation methods of the system are provided. The main aim of this paper is to tackle the problem of system calibration, a calibration method using traditional camera auxiliary is proposed, which can make full use of the existing open-source toolbox and only need to write a small amount of code to complete the calibration of the whole system.
The authors proposed an interesting method in this paper that can be useful for different applications.
The resolution of Figure 6 is not very well, and the two text boxes on the top are not easy to read.
In figure 5, the Depth map example is not provided, however, there is an example of a point cloud map.
The authors didn't mention the limitations of their work in the discussion section.
Figures 1, 2, 4, and 6 are not referred to in the manuscript. The authors should connect the figures to the text and refer to them in the manuscript.
There is not enough explanation about the Calibration of a catadioptric camera.
Author Response
Point 1: I The resolution of Figure 6 is not very well, and the two text boxes on the top are not easy to read.
Response 1: We have modified Figure 6 again to make it clearer on Page 13. The background of the text box was filled with white and the border was inside the image.
Point 2: The resolution of Figure 6 is not very well, and the two text boxes on the top are not easy to read.
Response 2: In Figure 5, we added a depth map to complete the flowchart on Page 11.
Point 3: The authors didn't mention the limitations of their work in the discussion section.
Response 3: Yes, we agree that the limitations of the work are not discussed in detail in the discussion in the manuscript. Combined with the opinions of other reviewers, we added why the reconstruction accuracy is not as good as that of the traditional camera 3D reconstruction system and how to improve it. One of the main reasons is the calibration error of the system. When calibrating a catadioptric camera, add pictures for calibration. The calibration error of the projector includes the calibration error of the traditional camera. We consider increasing the resolution of the projector to make the calibration error of the projector smaller. The second reason is that the distance of the reconstructed object is related. In the process of calibration and reconstruction, the focal length of the catadioptric camera is fixed. In order to shoot near the calibration plate, the focus is near. Then the further away the object is, the worse the reconstruction accuracy will be. The improvement method is to bring the reconstructed object closer to the catadioptric camera without affecting the projection of the projector and the calibration of the system. (Page17,Chapter 4)
Point 4: Figures 1, 2, 4, and 6 are not referred to in the manuscript. The authors should connect the figures to the text and refer to them in the manuscript.
Response 4: We fully agree with the reviewer that in the manuscript, there are no detailed quotes or descriptions of the images.We added sentences to the manuscript which are [In Figure 1, we can clearly see the process of projecting a point in the camera coordinate system to the image pixel ]. (Page2, Chapter 2.1.1, line 3)
Figure 2 displays the schematic diagram of the pinhole imaging model. It furthermore explains how the point in the camera coordinate system is projected to the physical image plane through the camera aperture, and finally the corresponding pixel coordinate is obtained. (Page 4, Chapter 2.1.2, line 3)
For Figure 4, in combination with comment # 5, the modification was made. (Page6-7, Chapter 2.2.2 line 1)
We’ve changed [The hardware of the experimental system is illustrated in Figure 6.] to [As shown in Figure 6, the hardware part of the experimental system consists of a catadioptric camera, a traditional camera, a projector, and a computer system. The left and right sides of the screen are respectively placed traditional camera and catadioptric camera while projector is placed against the screen.] (Page 12, Chapter 3.1, line 1)
Point 5: There is not enough explanation about the calibration of a catadioptric camera.
Response 5: Yes, we agree that The content on calibration of reflex camera is not complete.In view of the inadequate calibration of catadioptric camera, Chapter 2.2.2 "Calibration of Catadioptric Camera" has been added to the manuscript. In Chapter 2.2.2, two points are first clarified: one is that the reflector and the traditional camera should be calibrated as a whole; the other is that the checkerboard should be made large enough and close to the catadioptric camera.
In order to more vivid description of catadioptric camera calibration process, I added four pictures in Figure 5. The first is the initialization of parameters, including the distortion parameters, through Hough circle method detection circle center to calculate the main points, at least three points on the horizontal line to estimate the focal length. The catadioptric camera is calibrated according to the relationship between the 2D and 3D coordinates of the detected feature points. (Page 5, Chapter 2.2.2, line 1)

Reviewer 2 Report
The authors proposed a reconstruction pipeline using a catadioptric camera and projector, including a calibration method using a traditional camera. As the authors mentioned in the abstract, this work is an implementation for an application in a small field for 3D reconstruction. It is not easy to find significant technical contributions in the manuscript. Also, the comparisons with other techniques are insufficient to show the performance gain. Due to the limitation of the technical contributions, I consider that this work is not easy to be improved to qualify for publication in the Sensors journal.
Author Response
Point 1: As the authors mentioned in the abstract, this work is an implementation for an application in a small field for 3D reconstruction. It is not easy to find significant technical contributions in the manuscript.
Response 1: Thanks a lot for the reviewer’s comments. But I do not agree with the reviewer.
In the first sentence of the abstract, the traditional camera has a small field of view. But the reviewer thought that traditional cameras are useful in small areas of 3D reconstruction. The field of view of a camera represents the maximum range the camera can see, usually in terms of angles. The larger the field of view, the greater the range of observations. The traditional cameras has a field of view angle of less than 60 degrees. The catadioptric camera used in this paper can image in all directions, and its field angle is 360 degrees horizontally and 115 degrees vertically. You can see the full image taken by the catadioptric camera in Figure 4.
Point 2: It is not easy to find significant technical contributions in the manuscript. Also, the comparisons with other techniques are insufficient to show the performance gain.
Response 2: Thanks a lot for the reviewer’s comments. But I do not agree with the reviewer.
- There are not many existing research methods for the calibration of catadioptric systems based on surface structured light. Jia T [19] uses the Taylor Polynomial calibration toolbox to calibrate the catadioptric camera, and uses accurate guides and projected pattern to calibrate the projector. In Cordova-Esparza's method [35], Taylor polynomial calibration toolbox is also adopted to calibrate components in steps. The checkerboard calibration for three times is required, and each step involves the installation or disassembly of components. In this paper, the calibration method of catadioptric system is adopted, and the traditional camera is introduced. The parameters of catadioptric camera and projector are skillfully related. The procedure is simple, and there is no need to disassemble and install devices in the calibration process.
- In addition, all of them used Taylor polynomial model to calculate 3D reconstruction point coordinates. In this study, the equation derived by the method of triangulation is used to project the catadioptric image onto the unit sphere, which verifies the feasibility of the unified sphere model for 3D reconstruction.
Point 3: Also, the comparisons with other techniques are insufficient to show the performance gain.
Response 3: Thanks a lot for the reviewer’s comments. But I do not agree with the reviewer.
- In Table 4, By measuring 10mm reconstruction accuracy comparison. According to the measured values, I improved by 70.83% and 93% compared with Jia[19], Cesar-Cruz, [20], respectively. What's more, they project spatially encoded structured light, and their experiments measure planes. And our reconstruction object is two real objects.
- In addition combined with the opinions of other reviewers, we added why the reconstruction accuracy is not as good as that of the traditional camera 3D reconstruction system and how to improve it. One of the main reasons is the calibration error of the system. When calibrating a catadioptric camera, add pictures for calibration. The calibration error of the projector includes the calibration error of the traditional camera. We consider increasing the resolution of the projector to make the calibration error of the projector smaller. The second reason is that the distance of the reconstructed object is related. In the process of calibration and reconstruction, the focal length of the catadioptric camera is fixed. In order to shoot near the calibration plate, the focus is near. Then the further away the object is, the worse the reconstruction accuracy will be. The improvement method is to bring the reconstructed object closer to the catadioptric camera without affecting the projection of the projector and the calibration of the system. (Page17, Chapter 4)
- In order to echo the beginning and highlight the innovation, we added in the introduction “There is a large variety of structured light in 3D reconstruction using traditional cameras are De Bruijn sequence [21] and Speckle dots which based on triangulation. Some are the N-step phase shift method [22], multi-frequency heterodyne method [23] which based on phase height method. Nevertheless, there are few types of structured light used in 3D reconstruction or measurement of catadioptric system, such as stereo-point spatial coding [20], hourglass structured light [19].” (Page 2) AND “Because of the uniform sphere model used by the catadioptric camera, a 3D space point coordinate formula based on the uniform sphere model is derived in this study. Due to the reconstruction of the white reflective geometry. It also shows the measures taken to decode the effect of structured light before and after the effect. (Page 2)”

Reviewer 3 Report
Dear Authors,
the paper is interesting and clear.
Some comments about the paper:
The height reconstruction errors seem too high. Have you any effective workaround to remedy this problem? Arguing this aspect could improve the “Discussion” section of the paper.
Also, some other test cases or sensitivity analyses on some setup parameters could help verify the robustness of the described calibration procedure.
Author Response
Point 1: The height reconstruction errors seem too high. Have you any effective workaround to remedy this problem? Arguing this aspect could improve the “Discussion” section of the paper.
Response 1: We fully agree to the comment that there is a high reconstruction error. We added why the reconstruction accuracy is not as good as that of the traditional camera 3D reconstruction system and how to improve it. One of the main reasons is the calibration error of the system. When calibrating a catadioptric camera, add more pictures for calibration. The calibration error of the projector includes the calibration error of the traditional camera. We consider increasing the resolution of the projector to make the calibration error of the projector smaller. The second reason is that the distance of the reconstructed object is related. In the process of calibration and reconstruction, the focal length of the catadioptric camera is fixed. In order to shoot near the calibration plate, the focus is near. Then the further away the object is, the worse the reconstruction accuracy will be. The improvement method is to bring the reconstructed object closer to the catadioptric camera and projector without affecting the projection of the projector and the calibration of the system. (Page17 Chapter 4)
Point 2: The probability for a given country h to be in a class k should be the proportion of observations (households) in country h that belong to the income class k. On page 9, the first equation (it would be easier for the reader if the equation is numbered) is not exactly the proportion of people because the authors take the sum of the probability. The interpretation of the equation in not obvious. Normally, after estimating a mixture of regression model we have for each observation its estimated probabilities to be classified into the different classes identified. What is often done is to classify a given observation into the class where its estimated probability is higher. In many software this is also the method used that gives us the proportion of people in each of the classes. The authors should explain the equation on page 9 and how to interpret it. Alternatively, they may use the proportion approach which will make the interpretation easier.
Response 2: Yes, we agree that the lack of some pictures or parameter settings to prove the robustness of the experiment affects the readability of the article.
Chapter 2.2.2 "Calibration of Catadioptric Camera" has been added to manuscript. two points are first clarified: one is that the reflector and the traditional camera should be calibrated as a whole; the other is that the checkerboard should be made large enough and close to the catadioptric camera. In order to more vivid description of catadioptric camera calibration process, I added four pictures in Figure 4. A reprojection error image is added to evaluate the catadioptric calibration error. (Page17 Chapter 4)
Although we initially wanted to add more projection errors to traditional cameras and projectors. We intend to use a few lines of text to verify the robustness of the calibration. However, the calibration of traditional cameras and projectors has not been updated in recent years and has been thoroughly studied. So we didn't add a lot of text to describe it. We add that the reprojection error of projector in X and Y direction is 0.81 and 0.53, respectively. (Page 13 Chapter 3.1)

Round 2
Reviewer 1 Report
The authors improved the paper and answered all the comments satisfactory. The revised paper included all the comments provided by this reviewer.
Reviewer 2 Report
Thank you for your reply.